# Temporal Trends in Apparent Food Consumption in Bangladesh: A Joinpoint Regression Analysis of FAO’s Food Balance Sheet Data from 1961 to 2013

**DOI:** 10.3390/nu11081864

**Published:** 2019-08-10

**Authors:** Syed Mahfuz Al Hasan, Jennifer Saulam, Kanae Kanda, Tomohiro Hirao

**Affiliations:** 1Department of Public Health, Faculty of Medicine, Kagawa University, Kagawa 7610793, Japan; 2Department of Food Processing and Nutrition, Karnataka State Akkamahadevi Women’s University, Vijayapura, Karnataka 586108, India

**Keywords:** apparent dietary intake, food balance sheet, joinpoint regression analysis

## Abstract

We analyzed the temporal trends and significant changes in apparent food consumption or availabilityin Bangladesh from 1961 to 2013. Due to the lack of a long-term national dietary intake dataset, this study used data derived from the FAO’s food balance sheets. We used joinpoint regression analysis to identify significant changes in the temporal trends. The annual percent change (APC) was computed for each segment of the trends. Apparent intake of starchy roots, eggs, fish, vegetables, milk, and vegetable oils significantly has increased (*p* < 0.05) in the Bangladeshi diet since 1961; whereas cereals changed by merely 4.65%. Bangladesh has been experiencing three structural changes in their dietary history after the Liberation War, though the intake level has been grossly inadequate. Initially, since the late-1970s, apparent vegetable oils intake increased at a market rate (APC = 7.53). Subsequently, since the early-1990s, the real force behind the structural change in the diet has been the increasing trends in the apparent intake of fish (APC = 5.05), eggs (APC = 4.65), and meat (APC = 1.54). Lastly, since the early 2000s, apparent intakes of fruits (APC = 20.44), vegetables (APC = 10.58), and milk (APC = 3.55) increased significantly (*p* <0.05). This study result reveals and quantifies the significant secular changes in the dietary history of Bangladesh from 1961 to 2013. Bangladesh has experienced inadequate but significant structural changes in the diet in the late-1970s, early-1990s, and early-2000s. Overabundance of cereals and inadequate structural changes in the diet may have caused the increasing prevalence of overweightness and emergence of diet-related, non-communicable diseases in Bangladesh.

## 1. Introduction

The changes in dietary history have accelerated over the last three centuries and after the Second World War, it has gained the momentum to change more quickly. This quick dietary change can be linked to the worldwide emergence of diet-related non-communicable diseases [1], and the burden of these diet-related, non-communicable diseases (DR-NCDs) in developing countries is reported to be as much as 78.0% [2]. The changes in dietary intake patterns, both in terms of food and nutrient intakes in developing countries, are likely to compound the adverse effect of a nutrition transition [3]. This adverse transition results in the increasing and wider prevalence of obesity and non-communicable diseases in developing countries, together with malnutrition. Being a developing country, Bangladesh is not an exception to this transition. Though Bangladesh has experienced a sizable decline in childhood malnutrition and poverty [4], DR-NCDs have emerged as a significant public health apprehension. In terms of the number of lives lost due to ill-health, disability, and premature death, non-communicable diseases (inclusive of injuries) accounts for 61.0% of the total disease burden in Bangladesh [5].

Bangladesh has made a significant increase in food grain production and food availability, although food security at household and national levels remains a matter of major concern for the government. Improvement has been very slow in the quality and diversity of diet, which is a significant dimension of food security. The typical diet at the individual, as well as household level, in Bangladesh is not balanced and remains dominated by a high intake of cereals, mostly rice [6,7,8,9,10,11]. Considering the share of food groups, cereals intake, followed by non-leafy vegetables, roots, and tubers, together comprise more than four-fifths of the rural people‘s total diet [11]. Household income–expenditure surveys data suggest that per capita per day consumption of starches stayed stable and pulse consumption decreased markedly in the diet since 1985 [7]. Modest increases in the consumption of fish, eggs, vegetables, and spices have been reported since the 1980s [6,7]. The consumption of foods from different food groups have increased over time but it is grossly inadequate and far below the recommended level [7].

Tracing and monitoring the dietary intake at individual and population levels is required to assess the impact of dietary changes over time. Moreover, obtaining information on apparent and actual intakes is essential for predicting future dietary changes. This is supportive at the policy level to design protective measures and effective interventions for reducing the rising prevalence of DR-NCDs. Very few studies have reported the dietary changes in Bangladesh at a few points in time due to the lack of a long-term dietary intake database in Bangladesh [6,7]. A clear and more comprehensive picture of the dietary changes is crucial for understanding the patterns and causes of dietary changes in Bangladesh. In the case of Bangladesh, to study the temporal trends and dietary changes, the national data sources are very limited, and since 1973, only the Households (Income and) Expenditure Surveys (H(I)ES) from 1985, 1988, 1991, 1995, 2000, 2005, and 2010 are available [7]. Due to the lack of long-term, individual-level dietary data in many countries, most national information on food consumption is derived from food supply information, assembled in the food balance sheets. Food balance sheets provide a broad picture of the pattern of a country’s food availability during a particular reference period [12]. Moreover, this dataset helps to compare broad dietary trends over a very long timeframe and between countries. Food balance sheets do provide a good tentative picture of the overall food situation of a country, which is useful for economical and nutritional studies for developing plans and formulating projects [13]. Data on the national availability of the main food consumed within a country provide valuable information and insights into diets and their temporal evaluation over time. Hence, the aim of this study was to analyze the temporal trends and significant changes in food available for consumption or apparent food consumption in Bangladesh from 1961 to 2013. Moreover, we have also discussed some of the possible drivers that might have acted to change the apparent food consumption or availability in Bangladesh.

## 2. Methods 

### 2.1. Data Sources and Compilation

Like most countries, food supply derived from the food balance sheets is a reliable and perhaps a very good option available to follow and analyze the trends of dietary changes at the national level. In this study, due to the lack of a long-term national dietary intake dataset, food availability data in Bangladesh were obtained from the FAO’s food balance sheets documented in the Food and Agriculture Organization Corporate Statistical Database(FAOSTAT) from 1961, the baseline year, to 2013, the latest available year [14]. The food balance sheet data for Bangladesh was downloaded as comma separated values (.csv) files from the FAOSTAT database. The food balance sheet is compiled yearly and is an international resource of a country’s food availability during a certain period. Food availability is derived from production, supply, usages and wastages of food and provides information on the apparent consumption instead of actual consumption in the diet [15]. Assessing the pattern and trend in the availability of food components is a useful tool for the assessment of changes in diet and can be applied to define dietary patterns. Food availability in a country is calculated based on: the amount of food exported, used for animals and in agriculture, or wasted, and is deducted from the amount of food produced and imported; the annual amount of available energy from food items is then divided by the total population of a country in the same period [12]. Per capita, food availability data are represented as both kcal/day/person and kg/day/person in the FAOSTAT database [14]. For the food intake, we obtained the dataset from FAOSTAT of the fully adjusted food supply with kg per capita per year values for each food item. We then divided the values in kg per capita per year by 365.25 [16] to obtain values in kg per capita per day and multiplied this amount by 1000 to obtain g per capita per day for each food item. 

### 2.2. Operational Definition

Since the data came from national food balance sheets rather than from a nationwide dietary survey, these intake data refer to “average food and nutrients available for consumption.” This does not indicate the food actually consumed, rather it indicates national availability for food (g/day/person). Hence, in the remainder of this article “apparent food consumption or intake” should be read as “food available for consumption” or “national availability of food”[17,18,19].

### 2.3. Food Groups and Recommended Intake

For this study, dynamics and trends in apparent food intake or food availability were evaluated for eleven food items including cereals, starchy roots, pulses, fish, eggs, meat, vegetables, fruits, milk, vegetable oils, and sugar. Moreover, the food items that were grouped into these eleven food groups are presented in the Appendix A. In the meat group, the availability of meat and offal for consumption were added and combined in one group. We have not calculated the trends for animal fat intake because of its extremely low level of availability; from 1961 to 2013, animal fat intake was less than 1.0 g/day/person except in the year 1978 (about 1.01 g/day/person). Moreover, the nuts availability trend was also not analyzed for its extremely low availability (less than 3.0 g/day/person) in the diet. The recommended food intakes in the diet for the Bangladeshi population were derived from the desirable dietary pattern (DDP) for the Bangladeshi population [20]. 

### 2.4. Trends Analysis

To analyze the temporal trends and to identify significant changes in trends, we used Windows-based statistical software, the Joinpoint Regression Program (version 4.6.0.0, National Institute of Health, Bethesda, MD, United States), for performing the joinpoint regression by using joinpoint models [21]. This software tests whether an apparent change in a trend is statistically significant (*p* <0.05). With the joinpoint regression analysis, it is possible to identify years when a significant change in the linear slope of the trend is detected over the study period. The best fitting points, called “joinpoints,” are chosen when the rate changes significantly. The analysis starts with the minimum number of joinpoints and tests whether one or more joinpoints (in this study up to 5) are significant and must be added to the model. To describe linear trends by period, the estimated annual percent change (APC) is then computed for each of those trends. Moreover, we calculated the average annual percent change (AAPC) as a summary measure of the trends over the period for each of the food items. The average annual percent change calculated as a geometric weighted average of APCs of various segments [22] was used to quantify the trends of food intake changes in the diet of the Bangladeshi population over the entire available period of the FAOSTAT data from 1961 to 2013.

## 3. Results

### 3.1. Cereals and Starchy Roots

The apparent intake of cereals (g/day/person), the staple food in Bangladesh, showed a boat-shaped changing pattern in the Bangladeshi diet from 1961 to 2013 (Figure 1). During this period, cereals intake increased by only 4.65%, whereas a 4.4-fold striking increment was observed for the starchy roots apparent intake in the diet (Table 1). During the 1960s, cereals apparent intakein the diet significantly(*p* = 0.031) decreased by 1.68% per year (from 497.3 g/day/person in 1960 to 432.3 g/day/person in 1967), while annual apparent intake of starchy roots (dominated by potatoes) increased with a striking rate (APC = 10.32; from 29.6 g/day/person in 1960 to 57.6 g/day/person in 1969) during the 1960s (Table 2), especially during the late-1960s (Figure 1). After that, during the 1970s, 1980s, and up to the late-1990s, starchy roots apparent intake had been gradually decreasing by 1.71% per year (from 60.3 g/day/person in 1970 to 33.2 g/day/person in 1998), while the cereals apparent intake in the diet was almost stable with amarginal increase (APC = 0.19; from 461.2 g/day/person in 1968 to 474.4 g/day/person in 1997). During the late-1990s, the apparent intake of starchy roots reached almost half of the apparent intakelevelthat occurred in 1970. Since 1998, starchy roots intake significantly (*p* < 0.001) rose again with a striking rate (about 8.8% per year), similar to the 1960s trend. Unlike starchy roots, apparent cereals intake increased (APC = 2.40) for a very short time in the late-1990s, and after that, decreased at a marginal rate (about 0.25% per year) since 2001.

### 3.2. Fruits and Vegetables

Like cereals, apparent fruits (Figure 1D) intake showed a very similar boat-shaped changing pattern in the diet for the period from 1961 to 2013. Fruit apparent intake in the diet did not significantly change (AAPC = 0.2; 95% CI: −0.4 to 0.9; *p* = 0.50) during the last 53 years (Table 2) and increased by only 11.04% (Table 1). During the 1960s, apparent intake in the diet significantly increased (*p* = 0.003) by 2.48% (from 57.5 g/day/person in 1961 to 68.9 g/day/person in 1968) per year. Since then, apparent fruits intake decreased at a rapid rate (APC = −9.39) from the late-1960s to early-1970s and continued the declining trend at a comparatively slower rate (APC = −1.81) for the next three decades until 2002. During the period from 1968 to 2002, apparent fruits intake decreased by almost 2.4-fold. Since 2002, there had been a sharp and significant increase (*p* < 0.001) in the intake level with a striking rate of as much as 20.44% per year until 2007, and after that it decreased (APC = −0.73) but the slope was not found to be significant (*p* = 0.458).

Similar to fruits, a similar trend in apparent vegetable intake was observed during the last 53 years (Figure 1C). However, unlike fruits, vegetable apparent intake had changed (AAPC = 1.1; 95% CI: 0.5 to 1.7) significantly (*p*< 0.05) from 1961 to 2013 (Table 2), where it increased by almost 69.0% (Table 1). The apparent intake of vegetables in the diet had initially increased by 2.65% per year from 1961 (44.30 g/day/person) to 1968 (50.32 g/day/person), and thereafter it decreased by 4.68% per year up until the late-1970s (32.88 g/day/person in 1977). Since the late-1970s, for more than two decades, the apparent intake trend of vegetables in the diet was approximately stable (APC = 0.04) until 2001. Following this stable trend, there was a significant steep increase (*p* < 0.001) in apparent intake to as much as 10.58% per year during the 2000s and reached its peak in 2009 (from 33.70 g/day/person in 2001 to 72.64 g/day/person in 2009); after that, the increasing rate was drastically reduced and almost leveled off.

### 3.3. Meat and Pulses

Apparent intake of meat in the Bangladeshi diet did not change significantly over a 53-year period (AAPC = 0.3; 95% CI: −0.1 to 0.7; *p* = 0.10). Apparent meat intake in the diet increased by only 18.4% over this period (Table 1). During the 1960s and 1970s, the apparent intake was almost unchanged (APC = −0.09) in the Bangladeshi diet (Figure 2C), and thereafter, for a while the apparent intake was considerably diminished by as much as 10.22% per year from 1978 to 1981. Since 1981, apparent meat intake did increase at a slower rate of only 1.54% per year (Table 2).

From 1961 to 2013, pulses apparent intake in the diet increased by 43.5% (Table 1) (AAPC = 0.5; 95% CI: −0.9 to 1.8). During the 1960s, the apparent intake trend (APC = −0.73) was almost vertical (Table 2). During this time, apparent intake ranged from only 11.7 g/day/person to 12.5 g/day/person. After the late-1960s, sharp upward and downward trends were observed until 1983 (Figure 3C). The upward trend was characterized by a significant rapid increase (*p* < 0.001) in per capita apparent intake from 1968 to 1977 (APC = 6.86), which was followed by a similar sharp downward trend (APC = −6.27) until 1983. Since then, the apparent intake followed a declining trend with a significantly (*p* = 0.001) slower rate until 2008 (APC = −0.89) and back to the level that occurred in the 1960s. However, since 1996, apparent intake showed a variability that ranged from 7.69 g/day/person to 18.23 g/day/person. Since 2008, the slope of the intake trend was not significant (*p* = 0.09), though it had a sharp increase (APC = 6.29) and intake reached 17.98 g/day/person.

### 3.4. Fish

Apparent fish intake in the Bangladeshi diet increased by 2.25 times from 1961 to 2013 with an average annual increase rate of 1.6% (95% CI: 0.9 to 2.2) (Table 2). In the 1960s, 1970s, and 1980s, the apparent intake did not increase substantially and ranged from only 20 g/day/person to 33 g/day/person. From 1961 to 1973, a significant (*p* < 0.001) increase by 2.59% per year in apparent fish intake was noted, which was eventually followed by a drastic drop-off in intake level as high as 11.28% per year between 1973 and 1976 (Table 2). Between 1976 and 1991, apparent fish intake in the diet did not change significantly (APC = −0.55). Since the early-1990s, there was a marked shift in the apparent intake of fish in the Bangladeshi diet. From 1991 to 2010, there was a consequential sharp increase in apparent intake by 5.05% per year (*p* < 0.001) and the intake amount increased from about 21.0g/day/person to 53.0 g/day/person.

### 3.5. Eggs and Milk

For the last 53 years (1961–2013), the apparent intake of egg in the diet increased (AAPC = 3.4; 95% CI: 2.6 to 4.1) by as much as 6.5-fold but the intake level was grossly inadequate at only one-fifth of the recommended intake for Bangladeshi people (Figure 2B). During the 1960s, the apparent egg intake in the diet, though substantially very low, was markedly increased (from only 0.88 g/day/person in 1961 to only 1.75 g/day/person in 1968) by considerably 10.73% per year (Table 2). Belatedly, it reduced up until 1987 with a non-significant (*p* = 0.073) annual change of 0.95%, and subsequently, there was a significant increase (*p* < 0.001) in intake level by 4.65% per year from 1987 onwards.

Apparent milk intake showed a flat boat-shaped changing pattern in the diets of the Bangladeshi population for the period of 1961 to 2013 (Figure 2D). Apparent milk intake increased by 77.7% (Table 1) over this period (AAPC = 1.1; 95% CI: 0.6 to 1.6). In the 1960s, apparent milk intake was decreased by only 8.6% per year, but due to the variability in intake, the slope was not found to be significant (*p* = 0.406) (Table 2). Since the late-1960s, apparent intake started to increase in the diet. It showed an upward trend from 1967 to 2002 (APC = 0.70; only 31.1 g/day/person in 1967 to about 42.6 g/day/person in 2002), followed by a marked steep increase (*p* < 0.001) by 3.55% per year up until 2013 (about 60.0 g/day/person in 2013).

### 3.6. Vegetable Oils and Sugar

Apparent vegetable oils intake in the Bangladeshi diet increased by 2.60 times from 1961 to 2013 (AAPC = 1.9; 95% CI: 0.7 to 3.0). Initially, vegetable oil availability in the diet insignificantly decreased by 1.27% per year(*p* = 0.055) from 1961 (about 6.7 g/day/person) to 1978 (about 5.2 g/day/person), followed by a considerable annual rise by as much as 7.53% during the late-1970s to mid-1980s (about 9.3 g/day/person in 1985), and after that, continued to increase at a slower rate until 2000 (APC = 3.79) when they started to increase insignificantly(*p* = 0.368) by 0.88% per year (Table 2).

The apparent sugar intake in the diet did not change significantly (*p* = 0.50) over the 53 years considered (AAPC = −0.4; 95% CI: −1.5 to 0.7). During the 1960s, apparent sugar intake increased considerably by 8.13% per year (from 26.69 g/day/person in 1961 to 40.66 g/day/person in 1968) followed by a sharp significant decrease(*p* = 0.001)up until the early-1970s by as much as 9.26% per year (Table 2), and within a very short span with a decreasing trend, the apparent intake was back to the level that occurred in the early-1960s (Figure 3B). After the sharp decrease in the early 1970s, apparent sugar intake continued to decrease at a slower rate for three decades up until 2002 (APC = −1.40), followed by a non-significant (*p* = 0.212) rise up to 2005 (APC = 10.82), and again a significant decrease (*p* = 0.034) of 1.92% per year was noted for 2005 to 2013.

## 4. Discussion

The objective of the present study was to analyze the temporal trends and characterize the changes that have taken place in the apparent food intake of Bangladesh from 1961 to 2013 by using the FAO food balance sheet data. For the very first time, we have analyzed the food availability in the Bangladeshi diet for a period of 53 years. Moreover, this is the first paper that has reported the dynamics and temporal trends in the apparent dietary intake of Bangladesh in a comprehensive way by using joinpoint regression analysis over a period of 53 years, from 1961 to 2013. This is the first and longest historical food availability data analysis of Bangladesh.

Apparent intake of starchy roots, eggs, fish, vegetables, milk, and vegetable oils increased significantly in the Bangladeshi diet from 1961 to 2013. Cereals continued to remain by far the bulkiest food in the diet and did not change significantly since 1961. In the pre-Liberation War period, starchy roots, eggs, sugar, fish, vegetable, and fruit apparent consumption increased. The apparent intake of vegetables, fruits, and fish were seriously disturbed and disrupted before, during, and immediately following the Liberation War of 1971, and the country had been plagued by natural disasters almost without let-up from late 1970; only the pulses apparent intake increased over that time. After having long stable and downward trends, vegetables and fruits intake in the diet markedly increased during the 2000s. Since the mid-1970s, long-term stagnation or comparatively slow decreasing trends were observed in the apparent intakes of food items. Since the late-1970s, vegetable oils; since the late-1980s, fish, eggs and meat; since early-2000s, milk, vegetable, and fruit apparent intakes in the Bangladeshi started to increase significantly, though the increasing amounts have been inadequate compared to the recommended level of intake [20].

### 4.1. Trends in the Apparent Intake of Cereals and Starchy Roots

Cereals by far have been the most important food source in the world. In developing countries, mostly Africa and parts of Asia, cereals contribute to as much as 70% of the energy intake [23]. Bangladesh has traditionally had high cereals consumption. The commanding role of cereals, exclusively rice in the Bangladeshi diet, is clearly evident. The contribution of cereals to the dietary energy supply has declined slowly, reducing from 79.6% in 1995–96 to 77.0% in 2009–10 [24]. The apparent cereals intake showed a boat-shaped changing pattern in the diets of the Bangladeshi population, which is very similar to the trends of the energy and carbohydrate intake pattern. This structural pattern similarity revealed that the energy and carbohydrate intake of the Bangladeshi population was largely decided by the apparent consumption of cereals in the diet. This similarity also revealed the lack of sufficient diversity in the diet over the 53-year period.

During the 1960s, the cereals apparent intake in the diet significantly decreased by 1.68% per year, while the apparent intake of starchy roots (dominated by potatoes) increased (about a 2-fold increase) with a striking rate (10.32% annually) during the 1960s, especially during the late-1960s. Due to a significant breakthrough in agricultural progress during the 1960s, cereals production increased at a rate of 2.40% per year [25], and in another analysis, it was reported to be 3.17% per year from 1957/58 to 1970/71 [26]. The growth rate resulted from the introduction of improved cultural practice (solely Japanese method of line transplanting), the rapid expansion of multiple cropping, the introduction of modern irrigation facilities creating high-yielding seed varieties (HYV), and the popularization of chemical fertilizer [26]. Though production had increased in the 1960s, our analysis found a declining trend of per capita apparent cereals intake during that period (decreased by 1.68% per year). This phenomenon can be explained by the rapid rates of population growth (2.9% per year) during the 1960s [27]. While total net availability of cereals increased over that period, the increase was negated by the historically highest rapid rates of population growth [25]. Moreover, during the 1960s, Bangladesh experienced a secular decline in per capita availability of food and agricultural crops. This situation then resulted in the growing trend in food imports during the 1960s [26]. Unlike cereals, the production of starchy roots (solely potatoes) was among the highest and the annual production growth rate of potatoes was 12.0% [25], and in another analysis, it was reported that the growth rate was 11.86% per year from 1957/58 to 1970/71 [26]. The increase in production of potatoes was largely achieved by the introduction of higher-yielding varieties in the late-1960s [28], the introduction of modern irrigation facilities, and popularization of chemical fertilizers among farmers beginning from the mid-1960s [26].

From the late-1960s, the three consecutive decades of apparent cereals intake trend in the diet had been almost stable (only a 0.19% per year increment was calculated from 1967 to 1997) due to variability in the intake. During this period there were some positive and negative effects that affected the apparent intake and created high variability in apparent cereals intake. During the early-1970s, the production was disrupted due to the Liberation War and following natural disasters such as a cyclone (1970/71), drought (1973/74), and floods (1974/75) [26]. Moreover, before the mid-1980s, Bangladesh had experienced a near-stagnant situation in food grains production for two decades [29]. All these negative forces had reduced the apparent intake in the diet. Regarding the positive forces, development of groundwater irrigation by tube wells and rapid diffusion of shallow tube wells since the 1980s expanded the crop area and yield of dry-season *Boro* rice production noticeably [30]. Starchy roots intake was gradually decreased by 1.71% per year from 1969 to 1998 and reached almost half the apparent intake level of 1970. Since the late-1990s, apparent starchy roots intake significantly rose again with a striking rate of about 8.8% per annum, which is very similar to the 1960s trend. During the late-1990s, the production area of potato had been increased by 1.8 times (from 0.136 million hectares of land in 1997/98 to 0.245 million hectares of land in 1998/99), which resulted in an almost 80% increase in the production of potato [31]. Moreover, during this period, the local and high-yield varieties (HYV) potato production had increased markedly, and 61 modern varieties for potato developed by the national agriculture research systems in partnership with international agriculture research organization were released [32].

Like starchy roots, the apparent cereals intake increased by 2.40% per year but for a very short period of time in the late 1990s, and after that decreased at an insignificant marginal rate (about 0.25% per year). Though there were ups and downs in the production trends of food grains, in general, the production experienced an upward trend. The aggregate production of rice increased by 3.0% per year from 1980–81 to 2001–02. Moreover, this growth rate peaked to about 4.2% from 1996 to 2002 [29]. During this period, the rice-cropping pattern of Bangladesh changed and shifted from the rain-fed rice production to irrigation-based *Boro* cultivation [33,34]. Traditionally, *Aman* rice was the major source of rice in Bangladesh, but since 1998/99, the share of *Boro* rice superseded. Thus, Bangladesh had experienced a structural shift in rice production characterized by an irrigated crop rather than a mainly weather influenced rain-fed crop [35]. Now, *Boro* rice has a share of 55% in the total food grains production [34]. This structural shift by increasing *Boro* rice production resulted in the increasing trend of apparent cereals intake in the late-1990s.

### 4.2. Trends in the Apparent Consumption of Vegetables and Fruits

Consumption of fruits and vegetables plays a vital role in dietary diversity and providing micronutrients. Only a small and negligible minority of the world’s population consumes the generally recommended intake of fruits and vegetables [36,37]. In the context of Bangladesh, both the apparent and actual intake of fruits and vegetables have been inadequate and well below the recommended intake levels [7]. There were only 1.1-fold and 1.7-fold increases in the apparent intake of fruits and vegetable, respectively, in the diet from 1961 to 2013. During the 1960s, the level of fruits intake was comparatively good and apparent intake was significantly increased by 2.48% per year, whereas baseline vegetable intake was grossly inadequate but increased by 2.65% per year. The annual growth rate in production of vegetables during the 1960s was reported to be 5.9% [25]. Increase in the production of vegetables was achieved mainly by expanding the area for cultivation rather than using improved verities [25].

After the 1960s, the apparent intake of both fruits and vegetables were significantly reduced in the diet for three successive decades. The aftermath of the Liberation War, together with consecutive natural calamities during early-1970s, might have been an initial cause for these decreasing trends in apparent intakes. Moreover, a high emphasis has been placed on the intensive production of rice because of food sufficiency issues and facing a mono-crop situation [38], which might have decreased the production and hence the apparent intakes of fruits and vegetables. From the early-2000s, the per capita apparent intake of fruits and vegetables in the diet started to increase with a striking rate as high as 20.44% and 10.58% per year, respectively. In spite of the sharp increase in apparent fruit intake during the 2000s, the per capita apparent intake was up to the level that occurred in the late-1960s. Since the 2000s, fruit production in Bangladesh increased by 11.5% and Bangladesh became the 10thlargest country in the world in tropical fruit production [39]. Expanding the land area for fruit cultivation, commercial cultivation, and fruit trees plantation along roads and yards has caused a revolution in fruit farming in Bangladesh [39]. This revolution might have affected the increasing trend in apparent fruit intake in the diet since the 2000s.

Vegetable production in Bangladesh has increased since 1980, and up until 2003, the annual average growth rate was reported to be 2.8%, mainly due to the expansion in the area of production [40]. This increased growth rate did not translate into an increased apparent intake of vegetable since 1980. This was because, compared with rice, vegetables appeared to be highly competitive in terms of returns [41] and per hectare production of vegetables were more costly than traditional crops [40]. In contrast, in our analysis of the late-1970s onwards, the apparent vegetable intake was almost unchanged (APC = 0.04). Actually, since the early-2000s, the vegetable production of Bangladesh increased at a marked rate, making Bangladesh one of the fastest-growing vegetable producers in the world. The promising growth in vegetable production stems from farmers’ adoption of hybrid seeds, better technologies, policy support, home gardening, higher returns, and cultivation of off-season and all-season vegetables [42]. Moreover, some 142 varieties of vegetables of indigenous and exotic origin are grown in Bangladesh [42].

### 4.3. Trends in the Apparent Consumption of Pulses

In Asia, pulses are an important low-cost addition in the diet along with staples like rice and wheat. In Bangladesh, pulses are said to be the meat of the poor because of their low cost. Pulse availability in the Bangladeshi diet has increased by only 1.43-fold from 1961 to 2013. In the 1960s, apparent pulse intake was almost stable and ranged from only 11.7 to 12.5 g/day/person. During that time, the production increased at a rate of 2.2% per year, which was exceeded by population growth rate and significantly exceeded by the production of vegetables, potatoes, and edible oil [26]. Pulse production declined due to a fall in acreage because pulses were grown in the dry winter season and there was a trade-off of pulses production against the irrigated high-yielding rice due to rice’s higher profitability [26]. Surprisingly, there was an increasing trend in per capita apparent pulses intake between 1968 and 1976 by 6.5% per year in spite of the huge disruption from the Liberation War and natural calamities. Since 1976, apparent intake declined initially at a striking rate (6.27% per year) until 1983 and then at a slower rate (0.89% per year) until 2008. This long declining trend in apparent pulses intake was also observed globally. Globally, pulses have declined in consumption level, particularly in the developing countries; apparent consumption reduced from 30.14 g/day/person in 1963 to 16.44 g/day/person in 2003 [23]. Moreover, the consumption of pulses in developing countries almost stagnated and drastically declined in Asia and sub-Saharan Africa [43]. These declining trends in consumption resulted from changing consumer preferences, failure to promote production, and achieving self-sufficiency in cereals. Overall, pulses productions at the global level grew at 1.3% per year from 1980 to 2013 but this was segregated into two phases. In the first phase, there was almost a period of stagnation in the production of pulses during the1990s, and in the second phase, production sharply increased since 2005. The bulk of the increase in production came from developing countries where both area and yield growth (from a low base) contributed to the production [44].

### 4.4. Trends in the Apparent Consumption of Animal Products

There has been a considerable increase (about 62.0%) in the availability of meat worldwide, with the biggest increases in the developing countries [23]. Like meat, egg consumption has doubled worldwide and the increases more marked in the developing countries. Milk consumption has risen in some developing countries, especially in the Asian region from 38.4 g/day/person in 1963 to 82.2 g/day/person in 2003 [23]. World and developing countries’ statistics on per capita consumption of meat, eggs, and milk have been pulled up by a number of large countries like China and Brazil [43]. Having the statistics of worldwide and developing countries growth in the sector of livestock, Bangladesh is also experienced an increased growth rate in production of this sector but per capita availability was still grossly inadequate. Since 1960, apparent meat and milk intake increased by 18.4% and 77.7%, respectively, in the diet. Eggs’ apparent intake in the diet increased from 0.5 egg/month/person in 1961 to 3.0 eggs/month/person in 2013. Though the apparent intakes of animal products (meat, egg, and milk) have increased since 1960, the apparent intake was still marginal and extremely inadequate compared to the recommended intake level for Bangladeshi people [20]. During the 1960s and 1970s, the apparent intake of meat was almost stable. Since the early-1980s, apparent intake started to increase at a slower rate of only 1.54% per year. During this time, bovine contribution to the diet was almost stable, whereas small ruminants (sheep and goats) and poultry mainly attributed to this increasing trend of apparent meat intake. Apparent milk intake, though it decreased in the 1960s, showed an upward trend from 1967 to 2002, and after that, showed a significant marked steep increase by 3.55% per year. Apparent egg intake in the diet, though substantially very low, markedly increased during the 1960s. After that, for three consecutive decades, intake was almost stable with very few ups and downs. Since the late-1980s, there was a significant increase in apparent egg intake in the diet by 4.65% per year.

Livestock has been an important part of the agriculture sector of Bangladesh. Prior to independence, courtyard production of livestock was an integral part of the farming system, but commercial farming was limited. A comprehensive analysis of a 60-year period (1949–2008) [45] on the actual headcount and growth of livestock in Bangladesh reported that average annual growth in bovine population was 1.0%, and for small ruminants (sheep and goats) it was 5.2%. Moreover, the highest annual growth was observed in the poultry sector (7.4%), which might have contributed to the increasing trend of apparent meat intake in the Bangladeshi diet since the early-1980s. The growth in poultry population was mostly due to a collective effect of the government’s emphasis and actions of the non-governmental organizations (NGOs) to promote commercial poultry rearing, which involved mainly women [45]. Since 1990, milk production has increased by 5.2-fold, meat production has increased by 12.7-fold, and egg production has increased by 7.2-fold [32]. The growing demand for high-value animal protein, as well as rising income levels and urbanization, might have been the driving force [46] for the expansion of the livestock sector in Bangladesh, and hence the underlying cause for the increasing trend in the consumption of animal products. In the third and fourth five-year plans (FYPs), Government focused on creating new employment opportunities for women and the landless through poultry rearing and improved breeding, medicine, and feed for poultry industries, as well as an increased supply of commercial breed chicks through commercial farming [45], which might have been an underlying cause for a significant increase in apparent intake of eggs in the diet by 4.65% per annum since the late-1980s. Moreover, these policies [47] accelerated trends of apparent meat consumption in the diet by increasing the production and availability of poultry meat. Since 2002, apparent milk intake showed a marked increase by 3.55% per year. This marked increase might have been due to the Government starting to focus on breeding improvement through genetic upgrades along with the preservation of native breeds and increased fodder supply by intensively using available land [48].

### 4.5. Trends in the Apparent Consumption of Fish

Apparent fish intake in the Bangladeshi diet increased by 2.25 times from 1961 to 2013. In the 1960s, there was an increasing trend in apparent fish intake in the diet, though the available amount for intake was almost half of the recommended intake [20]. During that period, the annual growth rate of production was reported to be only 1.8% per year [25]. The annual growth rate of fish production was found to be the lowest and lagged behind the growth rate of other agricultural commodities. This poor production of fish was most probably due to limited supply in the inland waters, overfishing, and resource conflicts between fisheries and crop production [25]. Excessive use of poisonous pesticides, utilization of land for crop production, and irrigation making water not available for fishing were the major reported conflicts between fisheries and crop production that reduced the sufficient production of fish [25].

During the time of independence, fish cultivation was limited and most of the fish were caught from the inland water bodies. Moreover, the aftermath of the Liberation War, together with drought (1973/74) and floods (1974/75), had reduced the fish production and hence a drastic reduction (about 11.3% per year) in the apparent fish intake in the diet in the first half the 1970s. Due to this, in 1976, the per capita apparent fish intake was back to the level prevailing in the early-1960s. In the 1960s, 1970s, and 1980s, the apparent intake did not increase substantially. It was in the 1990s and afterwards when there was a marked shift in the apparent intake of fish and there was a consequential sharp increase in apparent intake by 5.05% per year and increased from about 21.0 g/day/person to 53.0 g/day/person. Usually, Bangladesh obtains fish from both inland and marine sources. At present, the major source of production in the country is inland fisheries. Between 1988/89 and 2012/13, total fish production increased by 4.1-fold. During this period, inland capture fisheries increased by 2.3-fold and inland culture fisheries have increased by as much as 10.1-fold, whereas production of marine fisheries has increased by 2.5 times [49]. One of the reasons behind the growth in the fisheries sector is that new farmers are coming out to produce fish, and both traditional rural aquaculture and intensive commercial aquaculture of high-value species of fishes have been widely produced. Moreover, some strict policy from the Government to stop fishing the mother fish with eggs and the small fishes increased the production to some extent [38]. In addition, in Asia, the production of aquaculture has been increasing due to the intensification of production methods. Small-scale traditional ponds, in which the range of carp species was stocked in a fertilized pond, has given way to farmed fish that is mostly reliant on affordable fish feeds [50]. Aquaculture produce has increasingly been an important component in the diets of the Asians. Furthermore, the continued and rapid expansion of aquaculture over the past 30 years has resulted in more than 40% of all fish now consumed being derived from farming [50].

### 4.6. Trends in the Apparent Consumption of Vegetable Oils and Sugar

Vegetable oils have experienced a high consumption growth in developing countries. This rapid consumption growth has been instrumental in increasing the apparent food consumption (kcal/day/person) of the developing countries [43]. This scenario has not been validated in the context of Bangladesh where cereals (especially rice) alone has contributed to more than 75% of apparent energy consumption. Starting from a low base (6.7 g/day/person in 1961) up until 2013, apparent vegetable oils intake in the Bangladeshi diet was inadequate, and in 2013, it was only 6.3% of the apparent energy intake in the diet. Apparent vegetable oils intake in the Bangladeshi diet increased by 2.60-fold from 1961 to 2013. During the 1960s and 1970s, apparent intake decreased by 1.27% per year followed by three phases of increasing trends. Since 1978, the apparent intake increased in the diet at a marked rate by 7.53% per year up until 1985 and then increased by 3.79% per year up until 2000. These marked increases in the diet were due to the increase in the import of vegetable oils since 1980. During this time, the import of vegetable oils grew at a much faster rate than the domestic production. Moreover, during the late-1980s, average import was about 2.6-fold higher than the domestic production, where it grew about 3.5-fold in the late-1990s. The marked increase in import had been due to the trade policy reforms by the Government with the objective of liberalizing the economy [38]. Eventually, during the 1980s, the trade liberalization progressed slowly, especially in the area of reducing import tariffs [51], but import tariffs, simplification of the trade procedure, and total tax incidence on the import of agricultural goods declined sharply during the early-1990s [52]. This favorable policy initiation and activation increased the import statistics of vegetable oils and hence the apparent consumption in the diet since the early-1980s. Moreover, a rapid growth of food demand, together with the rich calorie content of oil products, resulted in the increasing trends in apparent vegetable oils consumption in the diet [43].

Like vegetable oils, sugar consumption markedly increased among developing countries most notably in Asia, India, and in Latin America, and Africa in a lesser extent. However, inter-country and regional differences exist within the trends [23,43]. Bangladesh showed a declining trend in apparent sugar intake, unlike the developing countries. Apparent sugar intake in the diet did not change significantly over the 53 years considered. During the 1960s, the apparent sugar intake was more than the recommended sugar intake. More than 61.0% of the sugar was obtained from the non-centrifugal cane sugar. During that time, apparent intake increased sharply by 8.13% per year (from 26.69 g/day/person in 1961 to 40.66 g/day/person in 1968). This marked increase in the apparent intake was due to the increased intake of non-centrifugal cane sugar as sugarcane production grew at a rate as high as 7.0% per year from 1957/58 to 1970/71 [26]. Since 1968, apparent sugar intake reduced drastically (about 9.26% per year) and within a very short span was back to the level that occurred in the early-1960s. During the early-1970s, Bangladesh had experienced a serious lapse in the growth of cash crops production because of the Liberation War and successive crop failure caused by natural disasters. After the sharp decrease in the early 1970s, apparent sugar intake continued to decrease at a slower rate for three decades up until 2002. These long-term decreasing trends in the apparent intake was due to the reduced growth rate of sugarcane production. During that time, the expansion of cropped land under cereals had been partly at the expenses of cash crops, pulses, and oilseeds. For this reason, cash crops, jute, and sugarcane expansion was limited since independence. Sugarcane production grew at a rate of only 0.41% per year from 1970/71 to 1983/84 [26]. During the mid-2000s, there was a marked shift in the import of sugar, which increased the apparent intake for a very short time before starting to decrease again from 2005 to 2013 by 1.92% per year. Currently, only about 5.0% of the total sugar demand was produced in Bangladesh. About 20.0% of demand was fulfilled by non-centrifugal cane sugar production and the remaining total requirement was fulfilled by importation. The main causes of reduced production in the industry included a lower supply of sugarcane in the factories, very poor sugar recovery, and increased importation. The land under cane cultivation was drastically reduced due to the pressure of increased cereals production for attaining cereals self-sufficiency and other short-duration crops, which caused a lower amount of sugarcane production [53].

The summary of the significant changes in the diet is summarized in Figure 4. Overall, the apparent intake of starchy roots, eggs, fish, vegetables, milk, and vegetable oils increased significantly in the Bangladeshi diet from 1961 to 2013. Cereals continued to remain the bulkiest food in the diet by far and have not changed significantly since 1961. Per capita food use of cereals seems to have peaked in the late-1990s. Apparent food intake except cereals, though substantially inadequate, significantly increased during the 1960s. Starchy roots, eggs, sugar, fish, vegetable, and fruit apparent consumption increased during that time. Afterwards, the per capita apparent intake trends of food drastically went down due to the disruption of the Liberation War and subsequent natural calamities during the early-1970s. After this drastically declining trend, long-term stagnation or comparatively slow decreasing trends were observed in the apparent intakes of food items. Since the late-1970s, vegetable oils; since the late-1980s, fish, eggs, and meat; and since the early-2000s, milk, vegetable and fruit apparent intakes in the Bangladeshi diet started to increase significantly, though the increasing amounts were inadequate compared to the recommended level of intake.

The FAO’s food balance sheet data does have some major limitations that need to be described when describing the limitations of this study. Dietary data derived from this source does not indicate the actual consumption of food. This is an average quantity of food that was potentially available for human consumption. Therefore, it is not possible from our study to draw a conclusion on individual or sub-national level food intake patterns and it is infeasible to study inequalities. Other limitations of this source are some practical issues such as coverage and representativeness of the basic data because most of the statistics produced are confined to commercialized major food crops. Non-commercial or subsistence-level production, usually frequent in poor areas, is not included. There is a possible problem of overestimating food consumption since the food balance sheet does not take into account food losses that occur after the retail level. Food that is spoiled while processing at the household level, such as wasted trimmings, and as is common practice in rural Bangladesh, food fed to domestic animals within the households, are not accounted for in the calculation of food balance sheets. Another limitation is that there is some huge lack in detailing the fruits and vegetable categories in the food balance sheets. For example, for the vegetable categories, specific amounts are provided for tomatoes and onions, but all other vegetables are included in the section of “other vegetables,” which indicate the lack of sufficient information. Moreover, in Bangladesh, wild and indigenous types are the most consumed vegetables and contribute to most of the share of vegetable intake. For this reason, the statistics of apparent vegetable intake might be underestimated. The productions of both wild and indigenous vegetables are not taken into consideration in formulating the production statistics. Despite these limitations, the food balance sheet is the only cost-effective tool for temporal trend analysis and for longitudinal comparisons of dietary changes at a national level for Bangladesh. Moreover, it is still a very good available data source for the analysis of dietary changes during a specific period of time for a given country. In addition, with a wider view, food balance sheet data, together with dietary intake data and with a wide range of other evidence, can characterize the nutrition transition and its association with agricultural transformation and economic development [54].

## 5. Conclusions

This study allowed us to portray the temporal trends in apparent dietary intake in Bangladesh from 1961 to 2013. We found that the apparent intake of starchy roots, eggs, fish, vegetables, milk, and vegetable oils increased significantly in the Bangladeshi diet since 1961. Cereals availability in the diet was almost stable over the 53 years considered. Apparent food intake, though substantially inadequate (except cereals), significantly increased during the 1960s.The Liberation War and natural calamities during the early-1970s drastically reduced the apparent intake of fish, vegetable, fruits, and sugar. Bangladesh experienced three structural changes in its apparent dietary history after the Liberation War. Since the late-1970s, vegetable oils; since the late-1980s, fish, eggs, and meat; and since the early-2000s, milk, vegetables, and fruits available in the diet started to increase significantly, though the amounts have been grossly inadequate. These structural changes have increased the diversity in the diet, but the amount was grossly inadequate to have any positive effect on health. Most of these changes were related to the expansion effect and characterized by higher energy supply from cheaper foodstuffs of plant origins, mostly from cereals. The substitution effect, where shifts from carbohydrate rich staples to foodstuffs of animal origin at the same overall energy supply, started but the amount was grossly inadequate in the diet, even at national level availability in Bangladesh. In a nutshell, overabundance of cereals and grossly inadequate structural changes may act as the main causes of the increasing prevalence of overweightness and the emergence of diet-related, non-communicable diseases in Bangladesh.

## Figures and Tables

**Figure 1 nutrients-11-01864-f001:**
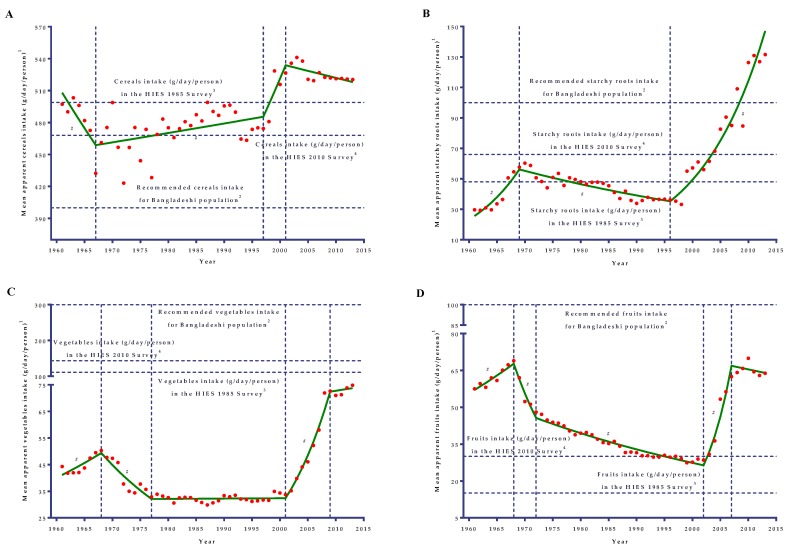
Joinpoint analyses of the mean apparent intakes of cereals (**A**), starchy roots (**B**), vegetables (**C**), and fruits (**D**) in the diets of the Bangladeshi population from 1961 to 2013. A vertical dotted line represents the joinpoints. ^1^ Mean apparent intakes of cereals (A), starchy roots (B), vegetables (C), and fruits (D) refer to the average availability of cereals (A), starchy roots (B), vegetables (C), and fruits (D) in the diets for consumption. ^2^ A horizontal dotted line represents the recommended cereals (A), starchy roots (B), vegetables (C), and fruits (D) intake based on the desirable dietary pattern (DDP) for Bangladeshi population [20]. ^3^ A horizontal dotted line represents the mean per capita intakes (g/day) in the diet of the Bangladeshi population based on the HIES 1985 survey [7]. ^4^ A horizontal dotted line represents the mean per capita intakes (g/day) in the diet based on the HIES 2010 survey [7]. # denotes the annual percent change (APC) that was significantly different from 0 (*p* < 0.05).

**Figure 2 nutrients-11-01864-f002:**
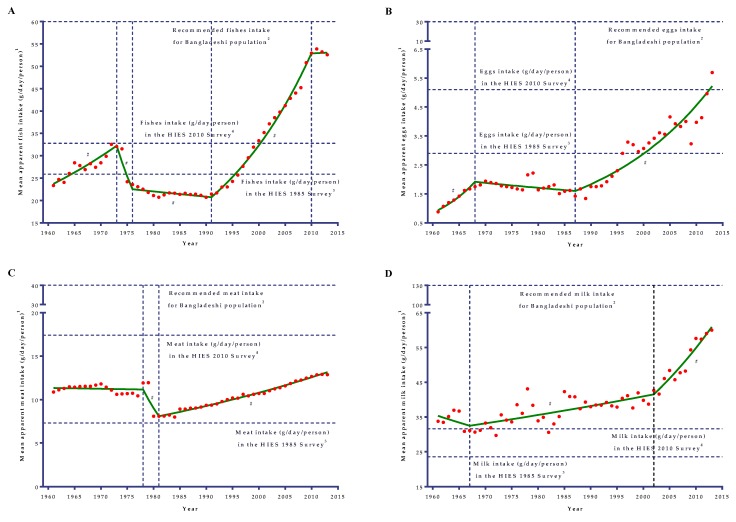
Joinpoint analyses of the mean apparent intakes of fish (**A**), egg (**B**), meat (**C**), and milk (**D**) in the diets of the Bangladeshi population from 1961 to 2013. A vertical dotted line represents the joinpoints. ^1^Mean apparent intakes of fish (A), egg (B), meat (C), and milk (D) refer to the average availability of fish (A), egg (B), meat (C), and milk (D) in the diets for consumption. ^2^A horizontal dotted line represents the recommended fish (A), egg (B), meat (C), and milk (D) intake based on the desirable dietary pattern (DDP) for the Bangladeshi population [20]. ^3^A horizontal dotted line represents the mean per capita intakes (g/day) in the diet of the Bangladeshi population based on the HIES 1985 survey [7]. ^4^A horizontal dotted line represents the mean per capita intakes (g/day) in the diet of the Bangladeshi population based on the HIES 2010 survey [7]. # denotes the annual percent change (APC) that was significantly different from 0 (*p* < 0.05).

**Figure 3 nutrients-11-01864-f003:**
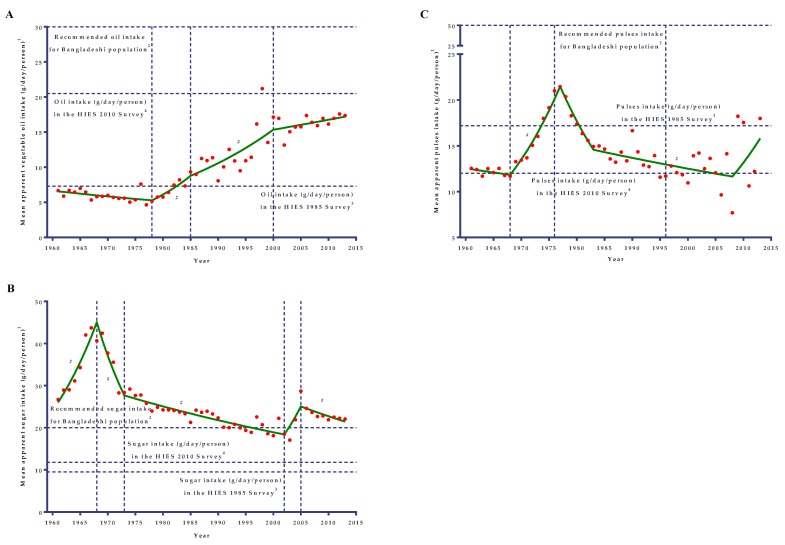
Joinpoint analyses of the mean apparent intakes of vegetable oils (**A**), sugar (**B**), and pulses (**C**) in the diets of the Bangladeshi population from 1961 to 2013. The vertical dotted line represents the joinpoints.^1^Mean apparent intakes of vegetable oils (A), sugar (B), and pulses (C) refer to the average availability of vegetable oils (A), sugar (B), and pulses (C) in the diets for consumption. ^2^A horizontal dotted line represents the recommended vegetable oils (A), sugar (B), and pulses (C) intake based on the desirable dietary pattern (DDP) for the Bangladeshi population [20]. ^3^A horizontal dotted line represents the mean per capita intakes (g/day) in the diet of the Bangladeshi population based on the HIES 1985 survey [7]. ^4^A horizontal dotted line represents the mean per capita intakes (g/day) in the diet based on the HIES 2010 survey [7]. # denotes the annual percent change (APC) that is significantly different from 0 (*p* < 0.05).

**Figure 4 nutrients-11-01864-f004:**
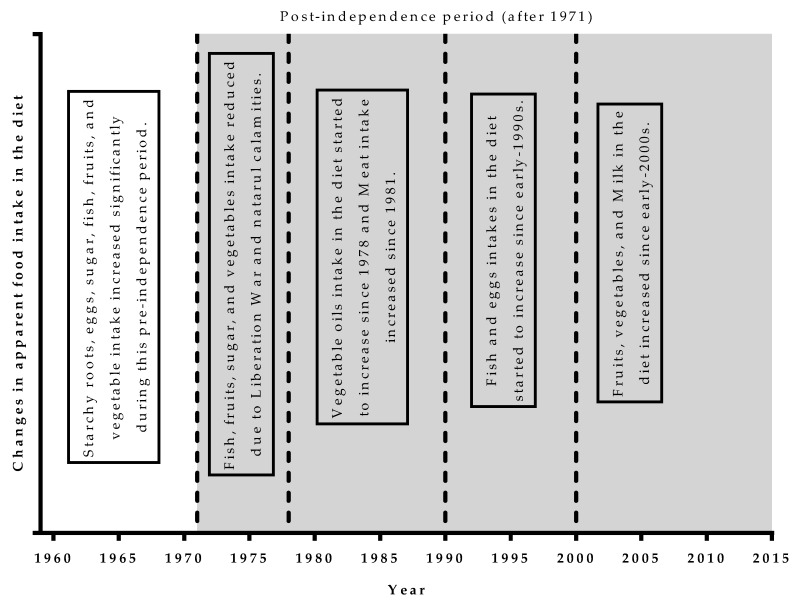
A brief summary of the significant changes over the periods in the apparent dietary intake in Bangladesh from1961 to 2013. Shaded area indicates the time after independence and the unshaded area represents the time before independence. The vertical lines represent the year where significant changes started.

**Table 1 nutrients-11-01864-t001:** Apparent consumption of food items in the Bangladeshi diet, 1961–2013.

Food Group	1961 (g/day/person)	2013 (g/day/person)	% Change^1^	Status^2^	*p*-Value^3^
Cereals	497.28	520.41	4.65	Increased	0.90
Starchy roots	29.60	131.53	344.36	Increased	<0.05
Pulses	12.54	17.99	43.46	Increased	0.50
Fish	23.35	52.59	125.23	Increased	<0.05
Eggs	0.88	5.69	546.59	Increased	<0.05
Meat	10.87	12.87	18.40	Increased	0.10
Vegetables	44.30	74.85	68.96	Increased	<0.05
Fruits	57.52	63.87	11.04	Increased	0.50
Milk	33.76	59.99	77.70	Increased	<0.05
Vegetable oil	6.68	17.36	159.88	Increased	<0.05
Animal fat	0.85	0.90	5.88	Increased	0.60
Sugar	26.69	22.07	17.31	Decreased	0.50

^1^ Values of the changes in intake level are given as an absolute figure; directions of the changes from 1961 are reported in the status. ^2^ Status has two responses: increased (when intake increased from the intake level of 1961) or decreased (when intake decreased from the intake level of 1961); the status is based on the direction of the percent change from 1961 to 2013, where 1961 is considered as a baseline year and 2013 is the latest available year. ^3^
*p*-Values are based on the average annual percent change of the different segment of the trend from 1961 to 2013 in each food item.

**Table 2 nutrients-11-01864-t002:** Trends in apparent food consumption in the diet in Bangladesh from 1961 to 2013 ^1^.

	AAPC ^3^	Trend 1		Trend 2		Trend 3		Trend 4		Trend 5	
	1961–2013	Period	APC ^2^	Period	APC ^2^	Period	APC ^2^	Period	APC ^2^	Period	APC ^2^
Cereals	0.0 (−0.4 to 0.4)	1961–1967	−1.68 *	1967–1997	0.19 *	1997–2001	2.40	2001–2013	−0.25		
Starchy roots	3.4 (2.8 to 4.1) **	1961–1969	10.32 *	1969–1998	−1.71 *	1998–2013	8.77 *				
Pulses	0.5 (−0.9 to 1.8)	1961–1968	−0.73	1968–1977	6.86 *	1977–1983	−6.27	1983–2008	−0.89 *	2008–2013	6.29
Fish	1.6 (0.9 to 2.2) **	1961–1973	2.59 *	1973–1976	−11.28 *	1976–1991	−0.55 *	1991–2010	5.05 *	2010–2013	0.08
Eggs	3.4 (2.6 to 4.1) **	1961–1968	10.73 *	1968–1987	−0.95	1987–2013	4.65 *				
Meat	0.3 (−0.1 to 0.7)	1961–1978	−0.09	1978–1981	−10.22 *	1981–2013	1.54 *				
Vegetables	1.1 (0.5 to 1.7) **	1961–1968	2.65 *	1968–1977	−4.68 *	1977–2001	0.04	2001–2009	10.58 *	2009–2013	0.45
Fruits	0.2 (−0.4 to 0.9)	1961–1968	2.48 *	1968–1972	−9.39 *	1972–2002	−1.81 *	2002–2007	20.44 *	2007–2013	−0.73
Milk	1.1 (0.6 to 1.6) **	1961–1967	−1.39	1967–2002	0.70 *	2002–2013	3.55 *				
Vegetable oil	1.9 (0.7 to 3.0) **	1961–1978	−1.27	1978–1985	7.53 *	1985–2000	3.79 *	2000–2013	0.88		
Sugar	−0.4 (−1.5 to 0.7)	1961–1968	8.13 *	1968–1973	−9.26 *	1973–2002	−1.40 *	2002–2005	10.82	2005–2013	−1.92 *

^1^ Trends analysis identified joinpoints, which are points where line segment of trends are joined. Each joinpoint denotes a statistically significant change (*p* = 0.05) in the trend. ^2^ APC is the annual percent change within a trend in food intakes from various food groups in the diet. ^3^ AAPC is the average annual percent change in food intakes from different food groups in the diet, calculated as a geometric weighted average of the calculated APCs of various segments from 1961 to 2013; the 95% confidence interval is presented in parenthesis. * Denotes that the annual percent change, APC, was significantly different from 0 for a specific trend (two-sided *p* < 0.05). ** Denotes that the average annual percent change, AAPC, was significantly different from 0 for the entire trend from 1961 to 2013 (two-sided *p* < 0.05).

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
