# Peer review of "Temporal Trends in Apparent Food Consumption in Bangladesh: A Joinpoint Regression Analysis of FAO’s Food Balance Sheet Data from 1961 to 2013"

_nutrients, 2019, doi:10.3390/nu11081864_

Round 1

Reviewer 1 Report

The authors insist using the term “apparent dietary intake” instead of country’s food availability/supply to define the dietary data analysed, which I consider a better description. As attested by them.

Looking for a better understanding why the authors proposed the term - “apparent dietary intake” I´ve made a search. I found a study addressing dietary data from Australian Aboriginal communities. They used store-turnover method to measure what they called apparent dietary intake, duet to their complex community food supply system. This term were used in a different approach from those proposed at this paper. That is why I strongly recommend to use country’s food availability/supply to describe the results of this study.

Authors should also revise:

- introduction exploring more the potential of country food availability data;

- their study goal: It could be better “to assess the pattern and trend in Bangladesh´s food availability”;

-Methods should indicate food items grouping and the sentence “the Joinpoint 115 Regression Program (Version 4.6.0.0), to analyse the trends by using joinpoint models.” Need to be reviewed due to repetition;

- Results are fragmented into food groups characteristics. Would be better to indicate general modifications to explore latter the main overtime differences according food groups;

- discussion section should present more arguments to explain possible drivers of changes in country´s dietary profile.

Sugested reference:

Lee, A. J., O'dea, K., & Mathews, J. D. (1994). Apparent dietary intake in remote Aboriginal communities. Australian journal of public health18(2), 190-197. 

Author Response

Comment 1: The authors insist using the term “apparent dietary intake” instead of country’s food availability/supply to define the dietary data analysed, which I consider a better description. As attested by them. Looking for a better understanding why the authors proposed the term - “apparent dietary intake” I´ve made a search. I found a study addressing dietary data from Australian Aboriginal communities. They used store-turnover method to measure what they called apparent dietary intake, duet to their complex community food supply system. This term were used in a different approach from those proposed at this paper. That is why I strongly recommend to use country’s food availability/supply to describe the results of this study. Reply 1: Dear Sir/Madam, thank you very much for your valuable comment and suggestions for our manuscript. We agree with you that “Apparent dietary intake” is not the appropriate term we used in our manuscript. We are sorry for not correcting the term in our previous version of the manuscript as you suggested. In our revised manuscript, we have omitted the term “apparent dietary intake” from title to the goal and wherever we use the term in our previous manuscript. Instead of this, we have used the term “apparent food consumption” or “food available for consumption”. The justification for this is that when we were conceptualizing our study and we started with the literature review in this filed to figure out what has been done and what scope we have, we came to know about this term. Moreover, when we created the road map of the articles and were making the annotated bibliography of the articles in this field then we found the term, “Apparent food consumption” but not the term “apparent dietary intake” to describe the result of our study. Hence we completely agree with you that “Apparent dietary intake” is a misleading term for our readers. In those literatures, we have found that “apparent food consumption” and “food available for consumption” and “food availability” are the terms that are synonymous to each other and are used in the articles to describe the results by analyzing the dataset of the Food Balance Sheets. [References: 17-19 in the revised manuscript]
1. Baldwin, K.; Schmidhuber, J.; Prakash, A.; Browning, J.; Kao, M.; Fabi, C. Guidelines for the compilation of Food Balance Sheets; Rome, Italy, 2017;
2. Alexandratos, N.; Bruinsma, J. World agriculture towards 2030/2050: the 2012 revision; ESA Working paper No. 12-03; Rome, Italy, 2012;
3. World Health Organization Diet, nutrition, and the prevention of chronic diseases: report of a joint WHO/FAO expert consultation; World Health Organization, 2003;

Comment 2:
- introduction exploring more the potential of country food availability data;
- their study goal: It could be better “to assess the pattern and trend in Bangladesh´s
food availability”;
-Methods should indicate food items grouping and the sentence “the Joinpoint 115
Regression Program (Version 4.6.0.0), to analyse the trends by using joinpoint models.”
Need to be reviewed due to repetition;
Reply 2:
- Introduction: In introduction we have explore more about the potentiality of food
availability data at national level with reference (Page 2 Lines No: 67-70 and 73-77)
- We have rewritten our study goal according to your suggestions and the literature we
have cited in support of the term used in our manuscript- “apparent food
consumption”
- Methods: In our revised manuscript we have added the food items grouping in details
in a supplementary Table (Supplementary Table S1). Yes Sir/Madam, there is a
repetition of the sentence content and we have rewritten it to overcome repetition.
Comment 3: - Results are fragmented into food groups characteristics. Would be better
to indicate general modifications to explore latter the main overtime differences
according food groups;
Reply 3: Thank you very much for your valuable suggestion regarding the structure of the
results. We would like to describe some of the background when we conceptualized the
structure of the results of our study. When we were discussing about the structure of the
results then we had two structures to describe our study results. One was to describe the
results where food groups would be the basis and we would describe the changes in
availability of each food group overtime and the second one was where timeframe would be
the basis and we would describe the food groups that changes over time on that preprescribed
time frame.
After discussion with our team, we decided to take food groups as the basis of our results
description. The justifications for choosing this structure are 1) it will help to visualize the
history of the per capita availability or apparent intake of each food items in a good way and
readers can easily know the history of changes and the drivers of these changes for each food
items. 2) On the other hand, if we would describe the result based on the timeframe then
some questions raised. How to define the fragmentation of the time frame over the 53 years?
Whether we would fragment the timeframe on 10 years basis or 15 years basis or 20 years
basis? If we select a timeframe then the next point is question about the consistency of the
changes overtime. The nodes of the joinpoint models were not same for each of the food
items’ with in a time frame. The joinpoints that we extracted from our analysis were not the
same for each food items and there is a variability of the joinpoints over the time frame for
each food items. 3) In this analysis, we have not selected the joinpoints over the time frame
by ourselves because this will make the study results very highly biased and which is also a
big problem for the segmented regression analysis. Through Monte Carlo simulation the
joinpoint models has created the joinpoints over time for each food group’s availability or
apparent consumption. So for all these reason and shortcomings, we have decided to describe
our results based on food group’s fragmentation. For the interest of the readers we have made
Table 4 to visualize the food group’s availability that have been changed over time but the
shortcomings and possible biasness to describe a uniform time frame we have chosen the
food group basis based description of our results.
Comment 4: - discussion section should present more arguments to explain possible
drivers of changes in country´s dietary profile.
Reply 4: Dear Sir/Madam. Thank you very much for your comment. In the discussion section
we have tried to present some of the possible drivers that acted to changes in the availability
of food at national level. Among the drivers we have discussed about the initiation of
agricultural technologies such as improved cultural practices, introduction of improved
irrigation facilities, popularization of chemical fertilizer, cropping pattern, and introduction of
high yielding varieties. In the agricultural drivers we have also discussed about the effect of
yield percentage, harvested area, and land use. We have also discussed about one of the main
drivers- the effect of population growth on the availability of food at national level. The
secular decline in per capita availability during the 1970s and 1980s was due to the stagnation
of production due to the natural calamities and disruption owing to the war of liberation. We
have discussed about this issues as the drivers of the stagnation or declining phase of the
availability of food at national level. We have discussed about the effect of the policy of the
government which affected the availability of food at national level such as trade
liberalization, liberalizing input markets, import, and tariff reduction policy. We have also
discussed about the Government five years plans that affected the per capita availability of
food at national level. In our discussion section we have tried our best to discuss about some
of the possible drivers of changes.
We have not discussed about some of the drivers such as income, economic growth, food
prices, urbanization, and effect of transactional food corporations, retailing process, food
industry marketing policies, and consumer’s attitude and behaviors. Because all these drivers
are directly and with a great extent related to the actual food consumption. These factor
works on the accessibility and actual consumption of food [1]. Thank you very much for
your valuable suggestion which improves our manuscript a lot. We are indebted to you.
1. Kearney, J. Food consumption trends and drivers. Philos. Trans. R. Soc. B Biol. Sci.2010, 365, 2793–2807.

Reviewer 2 Report

Thank you for addressing all of the comments.  Just do Grammar check once.

Author Response

Comment 1: Thank you for addressing all of the comments. Just do Grammar check once.
Reply 1: Dear Sir/Madam, thank you very much for your valuable comment and suggestions about our manuscript. Your valuable suggestions have helped us to improve the quality and scientific soundness of our manuscript. For grammatical mistakes we will contact with MDPI for professional English editing service for our manuscript as we do not have any native English speaking colleagues. Thank you very much again for your valuable comments, we are indebted to you.

This manuscript is a resubmission of an earlier submission. The following is a list of the peer review reports and author responses from that submission.

Round 1

Reviewer 1 Report

Thanks for this interesting manuskript in a highly important subjekt.  I would recommend you to check the language one more time. I also think table 1 would gain from a significant (p-value) for changes in each food item. 

Author Response

Dear Sensei

Reviewer 2 Report

General Comments and suggestions for the authors

This is an interesting manuscript which addresses trends in national food supply in Bangladesh, a country where direct food intake measures at national level did not exist for the analysed period. It offers a good opportunity to present a comprehensive over time picture of country food supply. It provides national data on food availability and can be used to assess groce estimations on food intake and nutrient profile. But, it is important to point out data limitations.

Considering the method used, the results can only account national level of the annual production of food, changes in stocks, imports and exports, and agricultural and industrial uses within the country. Dietary data derived from this source does not indicate the food actually consumed, but is an average of the quantity of food/nutrient (most of the analysis focuous on macronutrient profile) potentially available for human consumption. Thus, it is not possible to draw conclusions on individual or sub-national food consumption from these data and impossible to study inequalities. Other limitation of this source are some practical issues such as coverage/representativeness of the basic data, because most of statistics produced are confined to commercialized major food crops. Non-commercial or subsistence-level production usually frequent at poor areas are not included.

Over the manuscript the authors used different names to provide description of the same data. Eg.: ”apparent intake of egg in the diet increased “ , “Fish availability”. I recommend to use the term national/country avalilability as indicated by the manuscript authors ....“Since the data come from national food balance sheets rather than from a nationwide dietary  survey, these intake data refer to “average food and nutrients available for consumption”. ....Hence, in  the remainder of this article “apparent food intake” should be read as “food available for consumption”.

I would like to mention that with food balance sheet we can discuss dietary transition focusing on food production and economic issues related. I desagree with the authors statement “the food balance sheets is the most reliable and  perhaps the only option available to follow and analyze the trends of dietary transition.”. Considering this topic, I would like to recommend a reference that gave some arguments with a wider view (Masters et al., 2016). The research discuss the definition and the structural shifts in the historical relationship of economic development to diet-related disorders (the nutrition transition), food production and distribution (the agricultural transformation), and economic perspective.

Sugested reference:

Masters, W. A., Hall, A., Martinez, E. M., Shi, P., Singh, G., Webb, P., & Mozaffarian, D. (2016). The nutrition transition and agricultural transformation: a Preston curve approach. Agricultural economics47(S1), 97-114.

Author Response

Dear Sensei

Reviewer 3 Report

This novel study portrayed the temporal trends in apparent dietary intakes in Bangladesh from 1961 to 2013. Before it is considered for publication, I would ask the authors to make the suggested changes in the manuscript.

Page 1 – Lines 23-24: Rewrite the sentence as follows: Lastly, since the early 2000s, intakes of fruits……. and milk (APC=3.55) increased significantly (Also insert P value if available).

Page 1 – Lines 26-27: The concluding statement is quite general. You need to insert what are the implications of such structural changes in the diet on the health of the population

Page 1 – Line 31 - Replace quickened with accelerated

Page 1 – Line 32 – Replace force with momentum; Check Grammar

Page 1 – Line 39 – Replace declines with decline

Page 1 – Line 39-41 – Long sentence

Page 2 – Line 54 – over time but it is grossly……

Page 2 – Line 58-60: This is supportive at the policy level to design protective measures and effective interventions for reducing the rising prevalence of diet-related non-communicable diseases.

Page 2 – Lines 60-62: Check Grammar

Page 2 – Line 66: ‘a country’s food availability’

Page 3 – Line 95: ‘data came’

Methods – Did you use any software to analyse the data. Discuss it in the manuscript.

Page 3 – Line 119: Delete ‘and’

Page 3 – Line 127 – Delete ‘was’

Page 3 – Line 134 – Replace ‘a marginally increased rate’ with ‘a marginal increase’

Page 3 – Lines 135-136: Check sentence structure

Page 4 – Line159: Replace ‘was’ with ‘had’

Page 4 – The term ‘significantly’ has been used freuquently throughout the Results Section, but no P value has been mentioned. I would suggest inserting a P value.

Page 4 – Line 167: “…. Bangladeshi diet did not change significantly over..”

Page 4 – Line 175: Delete ‘was’

Page 5 – Line 178: Insert ‘was’ before followed

I have spotted a few Grammatical errors in the manuscript. Some of them have been highlighted above. Please revise them to improve the readability of the manuscript.

Line 352 – Reference needed for poor consumption of vegetables

Lines: 375-384: The increase in vegetable production has not translated to increased apparent intake. Any reasons?? Higher cost of fruits and vegetables or Westernisation of traditional diet etc.

Line 412: Reference for milk consumption in Asian region

Line 464: Reference required

With regards to import of vegetable oils, is Palm oil the most commonly imported vegetable oil?? If yes, please discuss it along with its health implications.

Also, discuss the impact of the structural changes in the diet over the years on the health of Bangladeshis. This will allow the readers to link your Introductory paragraph (discussion on non-communicable diseases) with the conclusion.

Author Response

Dear Sensei
